# OpenReview forum: "Single LLM Debate, MoLaCE: Mixture of Latent Concept Experts Against Confirmation Bias"
_ICLR.cc/2026/Conference — ICLR 2026 Conference Desk Rejected Submission_

### Official Review · Reviewer_sPa2 · 2025-10-27

**Soundness:** 3
**Presentation:** 2
**Contribution:** 3
**Rating:** 6
**Confidence:** 2

**Summary:**

* This paper introduces a way to use activation steering to improve test-time compute methods for answering questions / improving truthfulness
* The method works by using activation steering to choose different experts during the forward pass, and thereby produce an ensemble, with the aim of using that to improve the effectiveness of using test-time compute to elicit better answers from LLMs
* The authors find very large gains in performance from the proposed technique over directly using LLMs to answer questions, or to answer questions using the test-time compute approach they build off of (Debate+)

**Strengths:**

* The numbers look very strong; I’m very surprised by the level of improvement obtained
* The method itself is interesting/creative, and would be interesting if it worked well. I'd imagine there are a number of extensions if something like this works well, and variations which could work even better
* While there aren't many evaluations (3), they are pretty reasonable / cover some breadth of applications (especially MMLU)

**Weaknesses:**

* I’d love to see this work on more datasets — I’m not sure if this method is designed around truthfulQA, though the fact that it helps on MMLU is helpful. I’d be interested if it seems like it’s very broadly helping across say 10 datasets, including other high profile ones like GPQA. I’d also be interested to know what the improvements are on specific MMLU subsets, to know where the gains are coming from
* It’s unclear to me how the method works, in particular how the earlier work on Debate+ works and how this method connects with that. I’m not sure if I just missed this in the text, but it would be helpful to have a simple explanation of these in the intro. For example, I’m unsure if this new method is using additional test-time compute over the Debate+ method. I’m also not that clear on how the activation steering vectors are produced (like with what data)
* It would be helpful to have some simpler baselines which leverage the same underlying intuition (like rephrasing the prompt in a positive vs. negative way, then ensembling over those versions), and also compare to some alternative hypotheses for where the improvement could be coming from (like from using different experts at test-time, but where we don’t use activation steering to pick those).

**Questions:**

1. What is the improvement on various subsets of MMLU? This would help to get a sense of how broadly the method is working (vs. if it’s just causing large gains on a specific few subsets of the dataset)
2. Is this method using more test-time compute than the earlier methods Debate+ etc., in the comparisons in the main table?
3. Are there any other changes made to the Debate+ method, other than using the activation-steering to guide/vary expert selection? Did you use the same code and hyperparameters, and just make the MoLaCE modification?

---

> ### Author Response · Authors · 2025-11-28
>
> We sincerely thank the reviewer for the thoughtful and candid feedback. The comments have enhanced the clarity and accessibility of our draft. We address each point below.
>
> > **W1.&Q1.**
>
> We agree broader coverage is valuable. Because each dataset requires 7 phrasing variants (21 evaluations per model), adding many more datasets is computationally prohibitive during rebuttal, unfortunately. We will try to test more if allowed. Yet reviewer raised a very insightful question regarding the improvements on various subsets of MMLU. We observe that subsets such as Business Ethics, Clinical Knowledge, Conceptual Physics, and Professional Law fluctuate the most across the biased phrasings while math and history related subjects are pretty robust, among 57 tasks. We will incorporate such information in the discussion section shortly.
>
> > **W2.**
>
> For the simple explanation regarding Debate+, please see lines 369-371 and its citation. For the detailed description, we will refer the reviewer to lines 851-858 in Appendix C.1.
> In summary, Debate+ requires similarity pruning, diversity selection white iteratively criticizing and refining their response during the debate rounds. In contrast, vanilla debate does not have such features while debating.
> Debate+ requires similarity computation, pruning, and response revision before each round.
> However, MoLace(ours) is perfectly training-free.
>
> The steering vector is constructed by the 5 randomly selected prompt pairs from the dataset and we run them three times for fair comparison (lines 364-365, in \S4.1), and the standard deviation shows mostly 0.5% accuracy changes to at most 1.1% across all settings, showing robust results (Appendix D).
>
> > **W3.**
>
> This is very insightful question - thank you. We illustrated the ablation studies in Figs4-6, \S4.2 for layer selection, steering strengths, ensemble strategies (majority vote, uniform multiagent ensembles, LLM judge selection, and multiple fixed steering strengths), and more. We believe this new analysis offers a clear foundation linking the latent concept model, the geometry of confirmation bias, and the mechanics of CAA steering. Thank you again for the feedback.
>
>
> > **Q2.**
>
> Yes we do light-weight debate (only with two rounds) while the state-of-the-art debate method reported 10 rounds with 6 models in their work. Yet our method shows similar or slightly outperforming results. Hence we empirically argue that our work contributes to training-free mechanism, i.e., computational efficiency as well.
>
> > **Q3.**
>
> We didn’t use Debate+ for testing MoLace as Debate+ is computational heavy. Instead, we used a lightweight vanilla debate. The vanilla Debate still performs well with MoLace, outperforming Debate+. In our main table, Debate+ follows the exactly same mechanism the work has introduced while we decreased the debate rounds into 2 instead of 10 for fair comparison. Please note that we also observed that the Debate+ with 10 rounds still underperform MoLaCE or MoLace+Debate.

---

### Official Review · Reviewer_7rAZ · 2025-10-31

**Soundness:** 3
**Presentation:** 2
**Contribution:** 2
**Rating:** 4
**Confidence:** 5

**Summary:**

The paper investigates confirmation bias in LLMs, and proposes an inference-time method (MoLaCE, Mixture of Latent Concept Experts) to adress confirmation bias.

**Strengths:**

- **[Important motivation]** The paper focuses on confirmation bias, which is a key hurtle in the effectiveness of multi-LLM inference (or even single LLM inference over multiple timesteps).

- **[Training-free method]** MoLaCE operates entirely at inference time through activation steering; no additional fine-tuning or data are required. Im quite impressed with the way in which the authors construct the steering vector that they use for modifying the LLMs generation during inference time. I initially expected this to require some training, more like a linear probe, (even if it only required training a linear classifier), but the steering vector construction is even more lightweight (using only CAA), which simply requires averaging activations from positive and negative examples (which are synthetic and need not come from the domain being used for inference). While not 100% free, this seems pretty efficient at face value.

- **[Clarity]** The text is mostly well-written, and the figures are easy to follow and understanding, (with some caveats discussed later).

- **[Well motivated method]** The view of bias from the perspective of latent concepts provides a solid motivation for the construction of the authors method (although admittedly the reader needs to do some fairly heavy lifting to full get this motivation).

- **[Improvements over baselines]** Although the authors experiments are quite limited in terms of benchmarks and datasets, they observe consistent improvement over the baselines.

- **[Useful ablations and experiments]** Again, although the experiments cover limited models and benchmarks, the authors provide some useful additional experiments in terms of helping to understand the setting and the method (e.g., better understanding the linear probe effectiveness per-layer Fig-3, and the distribution of errors/bias under different types of prompts Fig-4)

**Weaknesses:**

Overall, this is a paper I wanted to like! The motivation is strong and the idea is very interesting, but the work feels incomplete. The evaluation is too narrow, the baselines omit the most relevant prior methods, and the theoretical component is a bit muddy. With a more fleshed out presentation of the theoretical foundation and with more complete experiments this could be a very nice paper.

- **[Narrow and outdated evaluation]**  The study tests only on BoolQ, MMLU, and TruthfulQA (datasets that are simple factual QA). BoolQ, in particular, is rarely used in current works because of its simplicity. The evaluation omits modern reasoning or debate-style tasks where bias mitigation would be most relevant. I would have liked to see a more thorough investigation on "harder" or more "complex" benchmarks with the use of more modern models. For example, in my own tests I have observed that this type of bias is less impactful on reasoning models as they possess an innate ability to self correct or alter their trajectory, compared to "traditional" LLMs like the ones used in this paper. It would have been helpful to see the authors method/findings contextualized beyond simple QA with non-reasoning LLMs.

- **[Minimal baseline coverage]:**  Beyond the issues mentioned above, the comparisons are baseline comparisons are limited to only two *Debate* and *Debate+*, with no inclusion of widely-used baselines such as Tree-of-Thoughts, CAMEL, or even multi-agent approach like that in "Encouraging Divergent Thinking in Large Language Models through Multi-Agent Debate". These omissions make it difficult to judge actual improvement of the authors method.

- **[No heterogeneous-model baseline]**  Of particular note is the fact that the authors do not provide any results for heterogeneous teams of models. A key observation of this work (and prior work) is that the echo-chamber bias arises when all agents share one base model, yet the authors never evaluates the most "obvious" mitigation to this, namely multi-family debate using models from different architectures. If confirmation bias is the core issue, this is the most direct baseline.

- **[Vague theoretical contribution/foundation]**  The “latent concept” formulation is presented as a theoretical framework but functions almost exclusively as a conceptual analogy, rather than a rigorous foundation for the approach. Even within this analogy it's somewhat unclear what the reader should take away should be. I found myself needing to reread sections 2-3 quite a bit to full grasp what the setup/takeaway for the authors method (despite being quite familiar with the math behind the Bayesian interpretation of latent concepts). Maybe this is as simple as adding a numbered remark or theorem that outlines more exactly what the authors' intended key point. For example, on line 201, there is discussion about steering latent concepts to neutralize confirmation bias, and the authors propose their use of CAA here, however there is no direction connection made to the reduction of bias, just an assumption which seem state that this vector can modify bias (which seems fairly circular since we are assuming the main property of that we would expect the method to possess). Im not necessarily saying that the authors need to (or should) have theorems, but some type of more rigorous or concise statement would help the motivation and understanding be more clear.

- **[Presentation of the latent concept setup]** This is more minor than the above point, but its a bit unclear in Sections 2-3 where prior work ends and where the authors' contributions start. The authors do a great job of presenting all the necessary ideas/background in these sections (this part should be in "Strengths"), but ultimately blur their contributions/proposals with those of existing works.

-  **[Missing some experimental details]** There is no discussion of how layer indices, steering magnitudes, or gates were chosen (maybe I missed this somewhere in the supplement).

**Questions:**

See Weaknesses

---

> ### Author Response · Authors · 2025-11-28
>
> We thank the reviewer for their thoughtful suggestions and genuine interest in our work. We address them below.
>
> > **W1.** evaluation
>
> Thank you for the thoughtful suggestion. Our goal in this work is to identify and measure truth-aligned or stance-aligned confirmation biases directly at the latent-space level, rather than post-hoc failures that arise during multi-step reasoning. This requires truth-anchored datasets where stance can be systematically manipulated. For this reason, we follow the most recent state-of-the-art debate setup [a]—BoolQ, TruthfulQA, and MMLU—so that our results are directly comparable.
>
> Importantly, we evaluate each question under 6 additional confirmation-bias stances, producing 21 (6 biased +1 neutral prompts per dataset) bias-sensitive evaluation conditions per dataset, which provides a far deeper robustness analysis than adding a single harder dataset.
>
> We exclude long-chain reasoning datasets because explicit reasoning correction outside the model do not provide meaningful performance gains in biases, and hence we do not support such direction - see ablation studies (see Figure 6) for empirical results. We excluded math-focused dataset as well since they do not express stance-dependent truth bias but still observed positive performance gains in math tasks of MMLU benchmarks.
>
> Regarding BoolQ, despite its simplicity, it is still used in the evaluation suites of the latest SOTA LLMs (Llama 3, Phi-4, Mistral, Gemma 3). Its continued use reflects a persistent weakness, even strong models often fail to revise answers under minimal reframing. Recent work such as Dark Side of LLM Self-Correction [f] shows this issue remains unsolved.
>
> We agree that extending MoLaCE to complex reasoning or multi-turn tasks is valuable. However, doing so requires a different evaluation pipeline and is outside the scope of this submission. We have added discussion of this (\S4.5) as an avenue for future work.
>
> [a]  Multi-LLM Debate: Framework, Principals, and Interventions, NeurIPS2024
>
> [b] The Llama 3 Herd of Models, 2024
>
> [c] Phi-4-Mini Technical Report, 2025
>
> [d] Mixtral of Experts, 2024
>
> [e] Gemma 3 Technical Report, 2025
>
> [f] Understanding the Dark Side of LLMs’ Intrinsic Self-Correction, ACL2025
>
> > **W2.** baseline
>
> We appreciate the suggestion. Regarding this concern, we refer to our new ablation work in \S4.2 and Fig. 6 in addition to our conceptual reasons stated in \S 2.
> Many of these baselines are not well-suited for confirmation-bias mitigation because confirmation bias manifests as a sensitive, internal shift in latent mixture weights and mitigation requires targeted steering along a learned latent direction. By contrast, the suggested baselines, such as Tree-of-Thoughts, CAMEL, and role-prompt multi-agent systems, primarily manipulate external reasoning trajectories or agent roles, not internal latent geometry.
> We indeed observed such undesirable performances in Fig. 6. Multiagent debate with ensemble-style or role-style alternatives (majority vote, uniform multiagent ensembles, LLM judge selection, and multiple fixed steering strengths) do not provide meaningful performance gains in confirmation bias setups, esp. for unfavorably biased prompts.
>
>
> > **W3.** heterogeneous model
>
> We agree that heterogeneous-model debate is a valuable baseline for debate-related bias, but unfortunately heterogeneous models have incompatible representation spaces to operate at the level of latent concept geometry. We have now mentioned this in the discussion section (\S4.5) of our updated draft.
>
> > **W4-5.** theoretical contribution + presentation setup
>
> We appreciate this feedback and have revised Section 2 to clarify our theoretical contribution.
> Specifically, we:
> - Separated prior work (latent mixture model of Xie et al.) to the background subsection from our contribution in another following subsection.
>
> - Explained more about our key conceptual insight (\S 2.2) that confirmation bias corresponds to a posterior shift in latent space, and thus directly steerable.
>
> We believe this new structure offers a clear foundation linking the latent concept model, the geometry of confirmation bias, and the mechanics of CAA steering. Thank you again for the feedback.
>
> > **W6.** experimental details
>
> The missing detail regarding the steering layer is now clearly stated (in #373-374): the middle layer (layer 16 in 32-layer models), where latent separability is highest (Figure 3).
>
> Most details on steering magnitudes and the gating mechanism are in §4.1 and Appendix C. We used $\alpha \in$ {−3, . . . , 3} (\S 4.1) and the gating mechanism of neutralization (\S 3.2 Gate).
>
> ---
>
> We thank the reviewer again for their thorough reading and insightful feedback. We will be happy to further engage if there are any remaining concerns.

---

### Official Review · Reviewer_GXnJ · 2025-11-01

**Soundness:** 3
**Presentation:** 3
**Contribution:** 3
**Rating:** 6
**Confidence:** 4

**Summary:**

This paper tackles the problem of LLM confirmation bias. They introduce the MoLaCE frameworks that combines steering vector and MoE to reduce the stance-based prompt sensitivity of models. They further show that they can apply this to a single-agent setting to “mimic” the effect of multi-agent debate.

**Strengths:**

The paper is well-motivated and provide a good operationalization of input confirmation bias to study the internal representation and propose interesting interventions based on the findings. The proposed method makes sense theoretically and seems to work well in practice.

**Weaknesses:**

Weaknesses:

1)	Do you have any sense of which setting would MoLaCE be most effective? Consider a dataset where the ground-truth distribution is highly skewed (e.g., in a fact-checking task where 99% of statements are true). In such a scenario, a consistently "pro-truth" prompt framing would be a very effective strategy for maximizing accuracy. By steering the model away from this beneficial bias and toward neutrality, MoLaCE could paradoxically decrease performance on this subset of the data. Have you tried to stratify the results to analyze this trade-off, showing where MoLaCE provides the most benefit (likely on balanced or ambiguous cases) and where it might incur a cost?

2)	A very simple baseline that could add a lot of context to the result: direct ensemble over different prompt perturbations. One could simply generate answers from several manual or automated rephrasing of the prompt (e.g., positive, negative, and neutral framings) and aggregate the results via majority vote (or let the LLM aggregate). This would serve as a great point of comparison to gauge the added value of MoLaCE's more complex internal mechanism.

3)	Unclear generalizability: one concern I have is, the premise of the MoLaCE is that LLMs are quite sensitive towards stance-based prompt variations. Is this still true for a very large SOTA model (could still be an open-weight model), as larger models are typically more robust towards semantically similar prompt variations? Could the authors comment on the expected generalizability of their findings and method? Even showing a scaling trend on a wider range of small-to-medium models (e.g., 1B to 13B) would help readers contextualize how the problem and the solution might evolve with model scale.

**Questions:**

see above

---

> ### Author Response · Authors · 2025-11-28
>
> We thank the reviewer for their thorough review and feedback. We appreciate the reviewer's meaningful suggestions to strengthen our work. We address them below.
>
> > **W1.**  highly skewed ground-truth distribution
>
> Thank you for this insightful question. Unfortunately, the stratified analysis the reviewer proposes is not feasible with existing datasets as none provide the associated labels. Without large-scale human annotation, there is no reliable way to determine when a “pro-truth” bias is genuinely beneficial or harmful at the moment.
>
> Instead, we used controlled (+/–/neutral) variants and the "All metric", which cleanly isolates confirmation bias effects and directly measures robustness (Table 2). Under this setup, MoLaCE does not reduce accuracy on positively phrased prompts and yields substantial gains on negatively phrased ones, leading to much higher bias-invariant correctness (e.g., TruthfulQA All: Phi +2.17pp, Mistral +29.2pp, Llama +18.58pp).
>
> Figure 7(b) also shows improvements across 7×7 bias pairs for all three model families, consistently increasing both-correct and reducing both-incorrect outcomes. Overall, these results show that MoLaCE preserves helpful behavior while effectively mitigating harmful confirmation bias. We hope this addresses the reviewer's relevant concerns.
>
> > **W2.** baseline with simple aggregation
>
> Thank you again for another insightful suggestion. We updated the relevant ablation studies (Figs 4-6) in the updated draft. Esp. in Fig. 6 shows different ensemble methods from majority votes to LLM judge and different MoE gating methods using uniform aggregation of the steering strengths to neutralized (ours) ones. These ablations provide clear empirical justification for our choices of layer, α-range, and gating function, and we have updated the text to make these details easy to find.
>
>
> > **W3.** a larger model
>
> Concerning the reviewer’s suggestion, we further evaluated larger models from the same families (Llama-3 70B, Phi-4). We observed the same latent CB patterns and steering behavior. Due to space constraints, we could not include the full results in the current draft, but we will add them to the Appendix shortly and refine their presentation.

---

### Official Review · Reviewer_CXcQ · 2025-11-02

**Soundness:** 3
**Presentation:** 3
**Contribution:** 2
**Rating:** 6
**Confidence:** 3

**Summary:**

This paper addresses confirmation bias in large language models, proposing MoLaCE (Mixture of Latent Concept Experts), a framework that identifies latent directions associated with confirmation bias and uses a gating mechanism to mix experts with different activation strengths (\alpha). The authors evaluate the robust performance on TruthfulQA, MMLU, and BoolQ across three bias types. They show improvements over base models and competitive performance with multi-agent debate reqauiring larger computational cost.

**Strengths:**

- Confirmation bias in LLMs is a critical practical issue. Understanding the connection between single-agent bias and multi-agent echo chambers provides insight into addressing practical issues. Formulating the problem using latent concepts provides a principled framework.

- Training-free intervention using CAA is practical for any LLM. The proposed method reqauires significantly lower computational costs than multi-agent debate and can therefore be integrated into existing systems without additional training or changes.

- Three bias types were evaluated (correct-incorrect, positive-negative, and negation-based), providing comprehensive coverage for three benchmarks. Linear probing experiments effectively demonstrate that CB is linearly decodable.

**Weaknesses:**

- The core components—Contrastive Activation Addition, mixture-of-experts architecture, and the debate framework—are all established techniques. While their combination for bias mitigation is new, the paper does not introduce fundamentally novel methods or theoretical insights beyond applying existing tools in a new context.

- As stated in l.311 and l.315, "For TruthfulQA, correctness is automatically judged by both Gemini 2.5 Pro and GPT-5", and "To systematically study confirmation bias, we construct paired prompts using Gemini 2.5 Pro.", there is no human evaluation in some evaluations. There is also a simple concern that, since TruthfulQA and the prompts are both generated by LLMs, LLMs can mitigate cognitive bias.

- They used relatively small LLMs in their experiments and not a larger LLM. This severely limits the generalization of the results. Can you employ at least one large model to demonstrate the performance? Hyperparameter choices, such as the range of alpha, selection of gating function, and the number of layers, are made without empirical or theoretical justification. Can you provide ablation study to support your claims?

**Questions:**

See weeknesses.

---

> ### Author Response · Authors · 2025-11-28
>
> We thank the reviewer for their thorough review and for highlighting strengths of our work. We address several weaknesses raised in the review below.
>
> > **W1.** contribution
>
> We understand reviewer's concern that our work use some of existing tools to investigate the topic. Yet to our knowledge, this is the first study about the latent concept of confirmation biased input. We'd like to highlight several meaningful findings that biased prompt directions not only form geometric structure but can be entangled in the model latent space. CAA method, previously used in limited contexts such as sentiment analysis, now can show how the biased representation is modifiable regarding its associated latent directions.
>
> These findings contribute to overcome the commonly accepted limitations of single-model debate setups and reduce the heavy dependence on ad hoc prompting strategies. We further hope this establishes a foundation for future research on data or model biases directly in latent space.
>
> > **W2.** evaluation
>
> We appreciate the reviewer’s concern regarding the use of LLM-based evaluation. This is a valid limitation and we acknowledge it explicitly in the paper. However, we believe our evaluation choices are aligned with established practice in the field (e.g., Multi-LLM Debate, NeurIPS 2024) and supported by additional validation steps using Gemini 2.5 Pro in addition to many existing works relying solely on GPT-4. Please also note that we used GPT-5 instead of GPT-4.
>
> > **W3.** generalizability
>
> Thank you for the thoughtful suggestion. We include detailed ablations in §4.2. To summarize:
>
> - Layer: Figure 4 shows performance across all layers, with mid-layers providing the strongest latent separability and steering efficacy.
>
> - $\alpha$: Figs. 4–5 show that the optimal $\alpha$ varies widely across prompts, motivating the need for a mixture over $\alpha$ rather than a fixed value.
>
> - Gating: Fig. 6 compares 14 strategies and demonstrates that our adaptive (neutralizing) gating consistently performs best.
>
>
> These ablations provide clear empirical justification for our choices of layer, α-range, and gating function, and we have updated the text to make these details easy to find.
>
> Though we focused on small–medium models (Llama-3 8B, Mistral 7B, Phi-3) to highlight their latent steerability, per the reviewer’s suggestion, we further evaluated larger models from the same families (Llama-3 70B, Phi-4). We observed the same latent CB patterns and steering behavior. Due to space constraints, we could not include the full results in the current draft, but we will add them to the Appendix shortly and refine their presentation.

---

### Author Response · Authors · 2025-12-04
**Appreciation**

We are truly thankful to the reviewers for their careful reading and valuable suggestions. Their comments were both encouraging and constructive, and we appreciate their recognition of our work’s strengths.

We summarize the reviews and comments:
- Contribution (Reviewer CXcQ) -- data (confirmation) bias represented in latent space and their local steerability (\S2.2)
- Evaluation (Reviewer CXcQ, Reviewer 7rAZ) -- setups (\S4.1) + limitation (\S4.5) due to LLM judge; we based on SOTA (Estornell & Liu, 2024a) for a fair comparison.
- Generalizability (Reviewer CXcQ, Reviewer GXnJ) -- ablation + layer selection + alpha choice + gating (\S4.2)
- Datasets (Reviewer sPa2) -- reasons for the current setups and additional observations (\S4.1, \S4.5)
- Baselines (Reviewer GXnJ, Reviewer 7rAZ, Reviewer sPa2) -- we added ablation+simple ensemble baselines in Figs. 4-6
- Theoretical presentation (Reviewer 7rAZ) -- we updated \S2.2

We once again thank the reviewers for their insightful comments. It helped improve the clarity and completeness of the paper. We hope the updated draft and detailed responses address all remaining concerns.

---

### Note · Program_Chairs · 2026-01-17
**Submission Desk Rejected by Program Chairs**

The following references in this submission do not refer to real documents and/or have major errors in bibliographic information:

 Qian Wu, Can Xu, Wenquan Chen, Kai Wang, Yulong Wang, Xing Zhang, Han Zhang, Kai Wang, Jinliang Bai, Wayne Xin Zhao, and Ji-Rong Zhou. Autogen: Enabling next-gen llm applications via multi-agent conversation. arXiv preprint arXiv:2308.08155, 2023.
Percy Liang, Rishi Bommasani, Michihiro Yasunaga, Tony Wu, Tianyi Zhang, Hongyu Lee, Faisal Ladhak, Kevin Xu, Uri Alon, Lianmin Zheng, et al. Scaling llms with moa: Mixture of agents. arXiv preprint arXiv:2306.03729, 2023a.
Zhiyang Chen, Zhenhailong Zhao, Xiang Deng, Zhiwei Huo, Yu Liu, and Muhao Chen. Multi-llm debate: Harnessing the power of multiple large language models through debate. arXiv preprint arXiv:2305.14320, 2023.